# Non-Invasive and Minimally Invasive Biomarkers for the Management of Eosinophilic Esophagitis beyond Peak Eosinophil Counts: Filling the Gap in Clinical Practice

**DOI:** 10.3390/diagnostics13172806

**Published:** 2023-08-30

**Authors:** Pierfrancesco Visaggi, Irene Solinas, Federica Baiano Svizzero, Andrea Bottari, Brigida Barberio, Greta Lorenzon, Matteo Ghisa, Daria Maniero, Elisa Marabotto, Massimo Bellini, Nicola de Bortoli, Edoardo V. Savarino

**Affiliations:** 1Gastroenterology Unit, Department of Translational Research and New Technologies in Medicine and Surgery, University of Pisa, 56125 Pisa, Italy; 2Division of Gastroenterology, Department of Surgery, Oncology and Gastroenterology, University of Padua, 35128 Padua, Italy; 3Gastroenterology Unit, Azienda Ospedale University of Padua, 35128 Padua, Italy; 4Division of Gastroenterology, Department of Internal Medicine, University of Genoa, 16132 Genoa, Italy

**Keywords:** biomarkers, eosinophilic esophagitis, ECP, EDN, MBP, EPO

## Abstract

Eosinophilic esophagitis (EoE) is a chronic esophageal disease that needs lifelong management and follow-up. The diagnosis requires an upper endoscopy with at least one esophageal biopsy demonstrating >15 eosinophils/high-power field, and often occurs with a diagnostic delay of up to ten years, partly due to the absence of valid non-invasive screening tools. In addition, serial upper endoscopies with esophageal biopsies are mandatory to assess the efficacy of any ongoing treatment in patients with EoE. These procedures are invasive, costly, and, when performed without sedation, are often poorly tolerated by patients. Therefore, there is the clinical need to identify reliable non-invasive or minimally invasive biomarkers that could be used to assess disease activity in clinical practice as a surrogate of peak eosinophil counts on esophageal biopsies. This review summarizes evidence on investigational non-invasive or minimally invasive biomarkers for the diagnosis and follow-up of EoE to report on the state of the art in the field and support future research. We discussed eosinophil-derived mediators including eosinophil cationic protein (ECP), eosinophil-derived neurotoxin (EDN, also known as eosinophil protein X), eosinophil peroxidase (EPO), and major basic protein (MBP) as well as other promising non-eosinophil-derived biomarkers. Although several studies have shown the utility of most biomarkers collected from the serum, esophageal luminal secretions, and feces of EoE patients, numerous limitations currently hamper the integration of such biomarkers in clinical practice. Future studies should aim at validating the utility of non-invasive and minimally invasive biomarkers using rigorous protocols and updated consensus criteria for EoE.

## 1. Introduction

Eosinophilic esophagitis (EoE) is a chronic, lifelong esophageal disease, characterized by symptoms of esophageal dysfunction and an eosinophil-predominant esophageal infiltrate [1]. Although the etiology is yet to be fully elucidated, the disease can be triggered by swallowed and inhaled antigens penetrating through a defective esophageal mucosal barrier [2,3]. In addition, it has been proposed that genetic predisposition, gastroesophageal reflux, and esophageal microbiota composition may have a role in the pathogenesis of EoE [3,4,5,6,7,8]. The diagnosis requires an esophagogastroduodenoscopy (EGDS) with at least one esophageal biopsy demonstrating >15 eosinophils/high-power field [1,9,10,11]. A diagnostic delay of up to ten years is common in EoE [12,13,14], with each additional year of undiagnosed EoE increasing the risk of strictures by 9% [15] as well as the risk of esophageal motility impairment [16,17,18]. Effective treatments for EoE are now available, with EoE-specific drugs ranking higher than older off-label drugs [19]. The epidemiology of EoE is rapidly evolving, and incidence and prevalence are rising at a rate that outpaces increased recognition [20,21]. A recent meta-analysis of population-based studies estimated a pooled prevalence of 34.4 cases per 100,000 inhabitants (42.2 for adults and 34.0 for children) and pooled incidence rates of 6.6/100,000 person-years in children and 7.7/100,000 person-years in adults [22]. In addition, it has been estimated that health care costs associated with EoE already largely exceed those of inflammatory bowel diseases and celiac disease, and are expected to increase further in the future [23,24].

Most EoE-related health care costs derive from the cumbersome follow-up of patients and assessment of disease activity [23]. In this regard, an EGDS with multiple esophageal biopsies is required for the diagnosis and assessment of treatment efficacy as well as maintenance of remission in patients with EoE [1,25,26]. This is particularly true for patients undergoing a six-food elimination diet, in whom up to seven EGDS in one year may be required to identify the appropriate dietary therapy [2,27]. These procedures are invasive, costly, and, when performed without sedation, are often poorly tolerated by patients, who may prefer less invasive tests [28]. Accordingly, there is a clear unmet clinical need to identify reliable non-invasive or minimally invasive EoE-associated biomarkers that could be used in clinical practice as a surrogate of peak eosinophil count on esophageal biopsies. In this regard, recent studies have individuated promising non-invasive EoE-associated biomarkers (Figure 1). The aim of this study was to review the evidence on investigational non-invasive or minimally invasive biomarkers for the diagnosis and follow-up of EoE to report on the state of the art in the field and support future research.

## 2. Major EoE-Associated Biomarkers

Eosinophils are pleiotropic leukocytes involved in the physiological immune defense against pathogens [29]. However, in the context of eosinophil-mediated diseases, they drive a chronic inflammation, leading to morphological changes of affected tissues as well as clinical symptoms [3,17]. The inflammatory activity of eosinophil granulocytes is mediated by highly cytotoxic proteins contained in eosin-stained specific granules located in their cytoplasm [29]. In particular, 90% of the content of eosinophil granules is represented by basic proteins including the eosinophil cationic protein (ECP), eosinophil-derived neurotoxin (EDN), eosinophil peroxidase (EPO), and major basic protein (MBP), which represent excellent markers of the presence and activation of eosinophils and have the potential to be used as biomarkers in eosinophil-mediated diseases such as eczema, allergic rhinitis, asthma, and food allergy [30,31,32,33]. It must be noted that, although the inflammatory infiltrate in EoE is eosinophil-predominant, several other cells and cytokines including mast cells, immunoglobulin (Ig) G4, periostin, and eotaxin-3 contribute to the inflammatory process of the esophagus in affected patients [3]. Major EoE-associated biomarkers and their possible utility in clinical practice are reported in Table 1.

## 3. Eosinophil Cationic Protein as a Biomarker

ECP is a single-chain, zinc-containing protein with a molecular weight ranging from 16 to 22 kilo Dalton (kDa) [31]. In mature eosinophils obtained from the peripheral blood of healthy subjects, ECP is localized in the matrix of eosinophil granules [33] and can be physiologically secreted outside of the cells in response to inflammatory stimuli. It has been shown that in patients with atopic diseases, eosinophils are more prone to secreting granule proteins upon stimulation [32,62]. Once secreted, ECP has potent cytotoxic activity thanks to its capacity to make pores in the cell membranes, leading to cell death due to osmotic lysis [63].

Several studies have investigated the potential role of ECP in the diagnosis and follow-up of EoE when assessed in the serum, on esophageal luminal secretions, and in stool specimens. In a prospective observational study conducted on 15 patients, Schlag and colleagues [36] investigated the utility of serum levels of ECP as a surrogate marker of response to topical steroid therapy in adults with EoE. The authors found a significant decrease in the mean ECP serum values under topical fluticasone treatment, and a concordant shift in serum ECP and eosinophilic count in 87% of patients, also paralleling a subjective improvement in symptoms, suggesting a possible role for serum ECP in the monitoring of EoE [36]. Similarly, in a randomized, double-blind, placebo-controlled clinical trial, the baseline and post-treatment serum ECP levels were assessed in patients undergoing a two-week treatment with orally disintegrating budesonide (BOT) or placebo [37]. Histological remission was achieved in 98% and 0% of patients randomized to BOT and placebo, respectively. Of note, the serum ECP levels significantly decreased in patients undergoing treatment with BOT, but not in patients taking placebo, who also failed to achieve histological remission. In another study, Witek and collaborators [38] showed that post-treatment serum ECP significantly correlated with the clinical response to elimination diets in patients with EoE, although the authors did not assess whether there was a correlation with histological remission. More recently, Cengiz and colleagues [39] investigated the accuracy of serum ECP in the diagnosis of EoE. The authors included 15 consecutive patients with a diagnosis of EoE and 14 healthy volunteers, and found that a cut-off value of 13.9 mcg/mL of serum ECP had 92.8% specificity and 80% sensitivity for the diagnosis of EoE. In addition, the study also showed that serum ECP significantly correlated with the severity of the EoE endoscopic reference score and a history of food impactions [39].

Most of the available studies on the use of serum ECP as a biomarker have included a small number of patients and were conducted prior to current consensus criteria for the diagnosis of EoE [9,10]. Therefore, further studies are needed to validate the role of serum ECP in the management of EoE.

In a recent randomized controlled trial, the levels of ECP identified with immunofluorescence on esophageal biopsies had excellent correlation with tissue eosinophil counts, showing a promising role of esophageal ECP in monitoring EoE [64]. In this regard, another study investigated the utility of ECP when assessed on esophageal luminal secretions collected with the esophageal string test (EST) [35]. The EST was developed in 2013 and consists of a minimally invasive string-based technology composed of a gelatin capsule containing a string. One end of the string is pulled from the capsule and wound around a finger. Subsequently, the capsule is swallowed by the patient and the proximal string is taped to the cheek of the patient, while the remaining string in the capsule deploys to end in the duodenal lumen, where the capsule is dislodged. When retrieved, the device allows the sampling of luminal secretions from the esophagus [28,35]. Furuta and colleagues [35] performed ESTs in 41 children with active EoE and control patients. Of note, EST measurement of ECP significantly distinguished between children with active EoE, treated EoE in remission, gastro-esophageal reflux disease (GERD), and normal esophagus, suggesting a possible utility of ECP measured with the EST for both the diagnosis and follow-up of EoE.

With regards to fecal testing, there is a lack of studies investigating normative concentrations of fecal ECP in both adults and children with eosinophilic gastrointestinal diseases (EGIDs) [65,66,67]. Nevertheless, some studies investigating the use of fecal ECP as an eosinophilic biomarker have shown promising results in both EoE and other EGIDs. A study conducted by Majamaa et al. [30] investigated fecal ECP in the monitoring of disease activity in pediatric patients with food allergy and atopic eczema undergoing an elimination diet. The authors compared fecal ECP trends in patients with or without confirmed milk allergy by collecting stool samples for ECP quantification at baseline and after three months of a rigorous elimination diet. The study demonstrated a distinct decrease in fecal ECP concentrations following a milk-free diet in most patients with confirmed milk allergy. Accordingly, the study concluded that fecal ECP could be a promising non-invasive tool in monitoring intestinal inflammation and disease activity in infants with eczema and food allergy. In another study, the same research group [34] investigated fecal ECP trends in infants and young children with atopic eczema with or without cow’s milk allergy undergoing a double-blind placebo-controlled oral cow’s milk challenge. The authors found that in patients with a cow’s milk allergy, fecal ECP concentrations increased following cow’s milk challenge. In contrast, in patients without a cow’s milk allergy, post-challenge fecal ECP concentrations showed no significant variations compared to prior to the cow’s milk challenge. With regard to fecal ECP in EoE, more recently, Ghisa et al. [40] showed that in a cohort of 29 EoE patients and 71 non-EoE controls, fecal ECP achieved 100% negative predictive value (NPV) for the diagnosis of EoE (diagnostic biomarker), discriminating all cases of active EoE from non-EoE controls, and 86% NPV for histological remission in patients with a diagnosis of EoE (treatment response biomarker). The authors showed a significant correlation between F-ECP and the histological activity of EoE (i.e., eosinophils/high-power field, HPF), correlation coefficient = 0.48, *p* = 0.008. Based on these promising results, further studies should aim at validating the performance of fecal ECP in the clinical management of EoE.

## 4. Eosinophil-Derived Neurotoxin as a Biomarker

EDN, also known as eosinophil protein X, is a neurotoxic protein of around 18 kDa that has 89% homology with ECP [68]. The name of EDN derives from its ability to induce a neurotoxic syndrome with ataxia and paralysis, known as the Gordon phenomenon, which originates in response to intracerebral injections of EDN [69,70]. Other than neurotoxicity, EDN alone has little to no cytotoxicity for somatic cells. However, when ligated to a single chain antibody against the transferrin receptor, EDN can be induce cytotoxicity due to its ribonuclease activity [71].

Dellon and colleagues [43] assessed the utility of serum EDN for the monitoring and follow-up of EoE in a prospective study on adult patients undergoing upper endoscopy with esophageal biopsies. Of note, there were no differences in serum EDN values between EoE patients and controls. In addition, EDN values did not change in response to treatment in treated EoE patients. In contrast, Konikoff and colleagues found that plasma EDN values correlated with tissue eosinophil levels and were increased in 47 children with active EoE compared to controls [41]. Similarly, in another study on 60 children with EoE and 20 control patients, Subbarao and collaborators found that serum EDN levels were significantly higher in subjects with EoE compared to controls, suggesting a possible role for serum EDN in the diagnosis of EoE [42]. In addition, the authors also found a significant decrease in the EDN serum levels following treatment, suggesting a potential role for EDN in the follow-up of children with EoE.

It has also been shown that marked tissue deposition of extracellular EDN is present in the esophagus of patients with EoE [72]. Accordingly, Furuta et al. [35] showed that the quantification of EDN measured in esophageal luminal secretions with the EST significantly distinguished between children with active EoE, treated EoE in remission, GERD, and healthy controls. More recently, Ackerman and colleagues [28] confirmed that EDN could be useful for monitoring disease activity in EoE, and showed that EST-collected EDN had an area under the curve (AUC) of 0.86 with 87% sensitivity and 71% specificity for segregating active from inactive disease. These results are consistent with those of Smadi et al. [44], who performed a blind esophageal brushing using a cytology brush inserted through a nasogastric tube. The authors included 94 children and young adults undergoing blind esophageal brushing, and compared the performance of brushing-collected EDN with peak eosinophil counts on esophageal biopsies as a reference standard. The study showed that the amount of EDN in esophageal brushings (bEDN) correlated with the peak eosinophil count in esophageal biopsies, and that bEDN had an AUC of 0.99 with a sensitivity of 98% and specificity of 89% for segregating active from inactive disease. In another study [45], bEDN collected during endoscopy was significantly higher in EoE patients nonresponsive to proton pump inhibitors compared to responsive patients, showing a possible utility of bEDN in predicting treatment response.

There are no studies investigating the accuracy of fecal EDN in the diagnosis or monitoring of EoE. However, at a mucosal and molecular level, EoE and food allergies share similar pathophysiological mechanisms [73], and studies on food allergies may be useful to inform further research on EoE. In this regard, Majamaa et al. [30] investigated fecal EDN levels in patients with food allergy and atopic dermatitis before and after an elimination diet and compared them with a population of non-atopic controls. Before the elimination diet, patients with food allergy had significantly greater levels of fecal EDN compared to the controls. Among the 19 atopic patients, 16 showed good response to the elimination diet and their fecal EDN levels decreased accordingly. One patient did not improve with diet, and their fecal EDN levels remained stable. In the study of Majamaa et al. [30], fecal EDN was proven to be useful in the monitoring of intestinal inflammation and the severity of the disease in patients with food allergy. Whether these results can be translated to the setting of EoE remains to be investigated.

## 5. Eosinophil Peroxidase as a Biomarker

EPO is a two-chain heme-binding protein with one heavy chain of about 52 kDa and one light chain of about 14 kDa [74], which is located in the matrix of the secretory granules and is specific to eosinophil granulocytes, since no other locations have been identified in mature cells [75]. The biological activity of EPO is mostly related to its peroxidase activity, which leads to the formation of long-acting radicals that damage cellular membranes [76,77].

With regard to serum EPO in EoE, Wright and colleagues [46] investigated the relationship between serum EPO and esophageal eosinophil counts in EoE and control patients. Although absolute median serum EPO did not differ between groups, when normalized for absolute eosinophil counts, EoE patients had significantly lower median EPO levels, showing an inverse correlation between serum EPO and esophageal eosinophilia in EoE [46].

It has also been shown that the EPO staining area at immunostaining on esophageal biopsies strongly correlates with tissue eosinophilia in EoE [47]. Accordingly, Wright and colleagues [47] showed that the assessment of the EPO staining area had high accuracy for the diagnosis of EoE, with an AUC of 0.95. In addition, the EPO staining area significantly decreased in treatment responders compared to non-responders. Similarly, Furuta and colleagues [35] confirmed the diagnostic accuracy of EPO assessed on esophageal luminal secretions when collected in a minimally invasive fashion with the EST. The measurement of EDN in esophageal luminal secretions accurately distinguished active EoE, treated EoE in remission, GERD, and healthy control patients.

Given the significant tissue deposition of EPO in active EoE patients compared to EoE in remission and controls, EPO could be a promising fecal biomarker in the management of EoE patients. However, studies on fecal EPO in patients with EoE are lacking.

## 6. Major Basic Protein as a Biomarker

MBP is a basic protein of 13.8 kDa that is produced as a much larger inactive protein named proMBP to protect cellular structures from the cytotoxic activity of MBP. Upon storage into the eosinophil granules, an acidic portion of the proMBP is cleaved off, and the active form of MBP is stored within eosinophil granules [74]. Although the exact mechanisms of MBP-related toxicity are still to be fully elucidated, it is known that MBP induces direct tissue damage. In addition, it has also been shown that MBP causes indirect cytotoxicity through the activation of other inflammatory pathways such as basophil-mediated histamine release, nonspecific complement activation, and neutrophil activation and degranulation [78].

With regard to the utility of MBP as a biomarker in EoE, Dellon and colleagues [43] found that there were no differences between EoE patients and controls in terms of serum MBP values at baseline. Moreover, MBP values remained stable post-treatment in the treated EoE patients. In contrast, Wechsler and colleagues [51] found that serum MBP predicted esophageal eosinophil counts in 183 specimens from 56 children with EoE and 15 non-EoE controls. In addition, when analyzing the patients with pre- and post-treatment specimens, the authors found that the MBP values decreased significantly in patients who achieved histological remission following treatment compared to active EoE.

Several studies have shown that there is a deposition of MBP in the esophagus of patients with EoE. Dellon and colleagues [48] compared the MBP density at immunohistochemistry in esophageal biopsy specimens of active EoE and non-EoE control patients. Of note, the MBP density was significantly higher in EoE and correlated with esophageal eosinophil counts. Accordingly, MBP had an AUC of 0.96 for the diagnosis of EoE [48]. Similar results were found in another study where the MBP density was significantly higher in EoE than controls [49], suggesting a possible role for MBP in the diagnosis of EoE. More recently, Peterson and colleagues [50] found that the MBP immunostaining grades related better to EoE symptoms than peak eosinophil counts. In contrast, Dellon et al. [79] showed that although the MBP levels at immunostaining were significantly higher in treatment non-responders compared to responders, the pre-treatment MBP levels did not predict treatment response. In accordance with studies investigating MBP immunostaining, Furuta and colleagues [35] showed the utility of MBP in esophageal luminal secretions collected with the EST. The authors found that MBP had an AUC of 0.97 for the diagnosis of EoE, and that EST-measured MBP could segregate between active EoE, treated EoE in remission, GERD, and healthy controls. More recently, Ackerman and colleagues confirmed that MBP measured with EST distinguished subjects with active EoE from inactive EoE or normal esophagi [28]. The authors also developed an EST-based score for the assessment of disease activity in EoE and found that the combination of MBP and eotaxin-3 distinguished active EoE from inactive EoE and normal controls with an AUC of 0.86 [28]. With regard to fecal MBP as a biomarker of EoE, to the best of our knowledge, there have been no studies on fecal MBP in EoE at the present time.

## 7. Other Non-Eosinophil-Derived Biomarkers in EoE

Although esophageal inflammation in EoE is predominantly eosinophil-driven, several other cells and cytokines are involved in the chronic esophageal inflammatory process [3]. A growing body of evidence is becoming available regarding the role of mast cells in EoE including participation in tissue inflammation [3] and symptoms perception [52]. Additionally, mast cells produce TGF-b, which stimulates muscle contraction and is involved in barrier dysfunction in EoE [3,16,17]. It has been shown that mast cell degranulation is high in patients with active EoE and can be elevated despite low levels of intra-esophageal eosinophils. Recently, elevated levels of MCT have been demonstrated on esophageal biopsies of patients with symptomatic EoE compared to asymptomatic esophageal eosinophilia, meeting the histological diagnostic threshold for EoE [52]. Similarly, Dellon and colleagues [49] showed that higher concentrations of MCT were present at immunostaining of esophageal biopsies in patients with EoE compared to non-EoE controls. Accordingly, as shown for eosinophil-derived mediators [28,35,44], the assessment of MCT in esophageal luminal secretions may be valuable in the setting of EoE, although no published studies on this are available yet.

Chronic food antigen exposure may lead to the development and deposition of IgG4s [80]. Accordingly, IgG4s are believed to be involved in EoE [3,53], although its specific role in esophageal inflammation is still unknown [80]. Serum concentrations of IgG4 have been shown to be higher in EoE compared to GERD and to decrease significantly following successful treatment of EoE [54]. Accordingly, serum IgG4s may be a potential biomarker of treatment response in patients with EoE. Similarly, esophageal local IgG4s have been shown to segregate EoE from GERD with a positive predictive value and negative predictive value of 92% and 80%, respectively [55]. Additionally, it has been shown that EoE treatment may have an effect on mucosal IgG4s, as a significant decrease in esophageal IgG4s has been observed following the successful treatment of EoE [54]. More recently, Pyne et al. [81] showed that patients with active EoE had elevated IgG2, IgG4, and IgM in esophageal luminal secretions compared to those with inactive EoE. In addition, the authors also found that food-specific IgG1, IgG2, IgG4, and IgM were significantly increased in patients with active EoE compared to EoE in remission.

Periostin is another player in the esophageal inflammation in patients with EoE and is involved in esophageal remodeling [3]. Periostin captured in esophageal luminal secretions with the EST showed correlation with eosinophil density, basal zone hyperplasia, and endoscopic appearance in EoE [56]. Similarly, eotaxin-3 levels significantly correlated with esophageal eosinophil density and were increased in patients with active EoE vs. controls [41] and in EoE vs. patients with GERD [48].

On a final note, the role of the microbiome in health and disease is being increasingly recognized [27,82], and alterations in salivary microbial ecology have been demonstrated in chronic inflammatory systemic diseases [58]. In this regard, saliva is easy and non-invasive to collect and offers an attractive biofluid for diagnosis and prognostic value and can be easily collected in patients with EoE. Accordingly, recent studies have shown alterations in esophageal and salivary microbiota associated with EoE as well as correlation with disease activity and a treatment effect on the microbiome. Initial studies demonstrated that the esophageal microbiota analyzed on esophageal biopsies or salivary samples of patients with active EoE was altered compared to EoE in remission or to patients without EoE [59,60]. In addition, more recently, Facchin et al. demonstrated that the analysis of esophageal microbiota on salivary samples allowed for segregation between active and inactive EoE [4]. Harris et al. [57] showed that the esophageal bacterial load was increased regardless of treatment status or degree of mucosal eosinophilia in patients with EoE compared to control patients. In contrast, Johnson et al. did not find any differences in the esophageal microbiome between EoE patients and non-EoE controls [61]. Therefore, the characterization of the salivary microbiome may be valuable in the management of EoE, but further prospective studies are needed to confirm these early results.

## 8. Conclusions

EoE is a chronic esophageal disease that requires lifelong treatment and follow-up for the monitoring of the safety and efficacy of any ongoing treatment [1,83]. Accordingly, to improve patients’ comfort and reduce costs, less invasive strategies of assessment of peak eosinophil counts such as unsedated transnasal endoscopy and Cytosponge™ have been proposed for the evaluation of esophageal eosinophilic infiltration [84,85,86,87]. However, it is well-known that symptoms and esophageal eosinophilia often show only modest correlation in patients with EoE [88]. Therefore, multiple esophageal biopsies to assess peak eosinophil counts are currently mandatory for the assessment of treatment efficacy regardless of the reported symptoms [10,88].

Recent randomized controlled trials have questioned the utility of peak eosinophil counts as a clinically meaningful endpoint in EoE [89,90]. In this regard, in two recent randomized controlled trials on novel biological drugs for EoE, namely the MESSINA (benralizumab) and KRYPTOS (lirentelimab) studies, most patients taking active treatment achieved histological remission, but the drugs failed to induce a significant improvement in symptoms compared to placebos [81,82]. In addition, a recent study comparing patients with symptomatic to patients with asymptomatic esophageal eosinophilia suggested that mast cells may contribute to the perception of symptoms in EoE, highlighting the possible role of other non-eosinophil-derived biomarkers in EoE [52]. Therefore, there is the clinical need to fill the gap between the currently accepted gold standard for the assessment of histological disease activity in EoE (i.e., peak eosinophil counts) and other relevant histological biomarkers of disease activity. For these reasons, research on EoE is increasingly focusing on the identification of non-invasive or minimally invasive biomarkers that could be used as a surrogate of or even replace peak eosinophil counts on esophageal biopsies.

Here, we performed a literature review to summarize evidence on potential non-invasive and minimally invasive biomarkers for the diagnosis and follow-up of EoE. We found that several studies have shown that biomarkers in the serum, esophageal luminal secretions, and feces could be useful in the management of patients with EoE, as shown in Table 1 and Figure 1. It must be acknowledged, however, that available studies are often limited by a non-randomized study design, a small number of included patients, and the use of outdated criteria for the diagnosis of EoE. We therefore stress that the biomarkers summarized in this review cannot be used in routine clinical practice yet. However, it is anticipated that some novel biomarkers, once validated, might possibly earn a relevant position in the management algorithm of EoE. Therefore, future studies should aim at validating investigational biomarkers using rigorous study protocols and updated consensus criteria for EoE. The validation of non-invasive and minimally invasive screening and follow-up strategies will optimize our capacity to diagnose EoE without delays and will improve the allocation of resources in the setting of EoE.

## Figures and Tables

**Figure 1 diagnostics-13-02806-f001:**
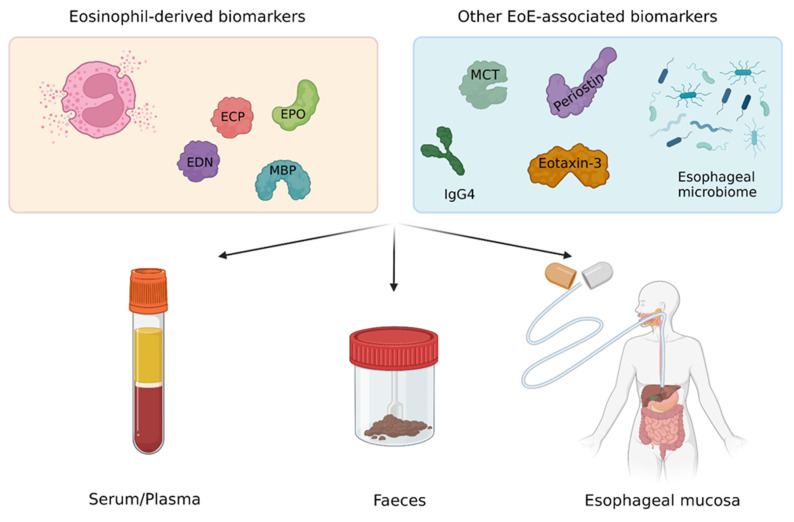
Major EoE-associated biomarkers and possible non-invasive or minimally invasive collection strategies. Abbreviations. EoE, eosinophilic esophagitis; ECP, eosinophilic cationic protein; EDN, eosinophil-derived neurotoxin; EPO, eosinophil peroxidase; MBP, major basic protein; MCT, mast cell tryptase; IgG4, immunoglobulin G4.

**Table 1 diagnostics-13-02806-t001:** Major investigational EoE-associated biomarkers in the serum, mucosa, and/or feces and their possible use in clinical practice.

Biomarker	Biological Specimen	Clinical Use	References
Serum	Mucosa	Feces
ECP	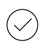	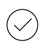	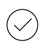	DiagnosisDisease activity biomarkerEndoscopic severity prediction	Majamaa 1996 [34]; Furuta 2013 [35]; Schlag 2014 [36]; Schlag 2015 [37]; Doménech 2017 [38]; Cengiz 2019 [39]; Ghisa 2020 [40].
EDN	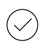	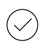	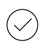	DiagnosisDisease activity biomarkerPredict treatment response	Majamaa 1999 [30]; Konikoff 2006 [41]; Subbarao 2011 [42]; Furuta 2013 [35]; Dellon 2015 [43]; Smadi 2018 [44]; Ackerman 2019 [28]; Irastorza 2022 [45].
EPO	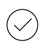	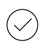	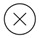	DiagnosisDisease activity biomarker	Furuta 2013 [35]; Wright 2018 [46]; Wright 2021 [47].
MBP	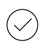	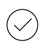	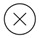	DiagnosisDisease activity biomarker	Dellon 2012 [48]; Furuta 2013 [35]; Dellon 2014 [49]; Dellon 2015 [43]; Ackerman 2019 [28]; Peterson 2019 [50]; Wechsler 2021 [51].
MCT	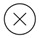	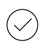	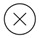	Diagnosis	Dellon 2014 [49]; Kanamori 2023 [52].
IgG4	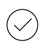	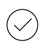	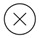	DiagnosisDisease activity biomarker	Clayton 2014 [53]; Weidlich 2020 [54]; Wong 2020 [55]
Eotaxin-3	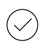	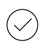	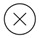	DiagnosisDisease activity biomarker	Konikoff 2006 [41]; Dellon 2012 [48]; Ackerman 2019 [28].
Periostin	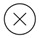	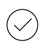	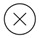	DiagnosisDisease activity biomarker	Muir 2022 [56].
Microbiome	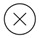	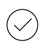	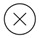	DiagnosisDisease activity biomarker	Harris 2015 [57]; Acharya 2017 [58]; Laserna-Mendieta 2021 [59]; Hiremath 2019 [60]; Johnson 2021 [61]; Facchin 2022 [4]; Massimino 2023 [5].

Abbreviations. ECP, eosinophilic cationic protein; EDN, eosinophil-derived neurotoxin; EPO, eosinophil peroxidase; MBP, major basic protein; MCT, mast cell tryptase; IgG4, immunoglobulin G4. 
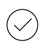
 indicates that the biomarker has been investigated in the corresponding biological specimen. 
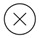
 indicates that the biomarker has not been investigated in the corresponding biological specimen.

## Data Availability

No additional data available.

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
