# Peer review of "Non-Invasive and Minimally Invasive Biomarkers for the Management of Eosinophilic Esophagitis beyond Peak Eosinophil Counts: Filling the Gap in Clinical Practice"

_diagnostics, 2023, doi:10.3390/diagnostics13172806_

Round 1
Reviewer 1 Report
The authors wrote a nice review on biomarkers in EoE, going in depth on previous studies evaluate serum, esophageal, and fecal biomarkers.
Comments:
- While I agree that minimally invasive biomarkers would advance the field, the authors state in the abstract and introduction EGDS are "often poorly tolerated" which is not universally accepted. The authors should point out that adverse effects of EGDS are considered rare both in adult and pediatric literature and cite this. In addition the adverse events are often related to anesthesia and not necessarily EGD.
- The authors do a nice job of outlining each biomarker with data from serum, esophageal eluate and feces.
- The authors should consider including a paragraph / section about other less invasive modalities including cytosponge, blind esophageal brushing, esophageal string test, unsedated transnasal endoscopy.
Katzka DA, Smyrk TC, Alexander JA, Geno DM, Beitia RA, Chang AO, Shaheen NJ, Fitzgerald RC, Dellon ES. Accuracy and Safety of the Cytosponge for Assessing Histologic Activity in Eosinophilic Esophagitis: A Two-Center Study. Am J Gastroenterol. 2017 Oct;112(10):1538-1544. doi: 10.1038/ajg.2017.244. Epub 2017 Aug 15. Erratum in: Am J Gastroenterol. 2017 Dec 19;: PMID: 28809387; PMCID: PMC5848207.
Katzka DA, Geno DM, Ravi A, Smyrk TC, Lao-Sirieix P, Miremadi A, Debiram I, O'Donovan M, Kita H, Kephart GM, Kryzer LA, Camilleri M, Alexander JA, Fitzgerald RC. Accuracy, safety, and tolerability of tissue collection by Cytosponge vs endoscopy for evaluation of eosinophilic esophagitis. Clin Gastroenterol Hepatol. 2015 Jan;13(1):77-83.e2. doi: 10.1016/j.cgh.2014.06.026. Epub 2014 Jul 3. Erratum in: Clin Gastroenterol Hepatol. 2015 Aug;13(8):1552. Miramedi, Ahmed [corrected to Miremadi, Ahmed]. PMID: 24997328.
Smadi Y, Deb C, Bornstein J, Safder S, Horvath K, Mehta D. Blind esophageal brushing offers a safe and accurate method to monitor inflammation in children and young adults with eosinophilic esophagitis. Dis Esophagus. 2018 Dec 1;31(12). doi: 10.1093/dote/doy056. PMID: 29905784.
Nguyen N, Lavery WJ, Capocelli KE, Smith C, DeBoer EM, Deterding R, Prager JD, Leinwand K, Kobak GE, Kramer RE, Menard-Katcher C, Furuta GT, Atkins D, Fleischer D, Greenhawt M, Friedlander JA. Transnasal Endoscopy in Unsedated Children With Eosinophilic Esophagitis Using Virtual Reality Video Goggles. Clin Gastroenterol Hepatol. 2019 Nov;17(12):2455-2462. doi: 10.1016/j.cgh.2019.01.023. Epub 2019 Jan 29. PMID: 30708107; PMCID: PMC6663663.
Philpott H, Nandurkar S, Royce SG, Gibson PR. Ultrathin unsedated transnasal gastroscopy in monitoring eosinophilic esophagitis. J Gastroenterol Hepatol. 2016 Mar;31(3):590-4. doi: 10.1111/jgh.13173. PMID: 26426817.
- The authors review each biomarker obtained from the esophageal string test under each heading, but they should include overall description of the esophageal string test with EoE Score (which specifically uses eotaxin-3 and major basic protein-1 concentrations) that distinguished subjects with active EoE from inactive EoE or normal. The EST is now being used clinically and not for research purposes only at several institutions and this should be noted.
- The biomarkers being reviewed overall alone are not being used in clinical practice as noted, but the authors should highlight less invasive modalities that are being used (Esophageal string test and transnasal endoscopy).
- In Section 7, line 301, the authors delve into recent literature on IgG4. This should be expanded to why IgG4 is believed to be involved in EoE so the reader understands why this is being evaluated as a biomarker.
- In line 319, the authors discuss the microbiome. They do not discuss the studies that show lack of association of microbiome and EoE.
Johnson J, Dellon ES, McCoy AN, Sun S, Jensen ET, Fodor AA, Keku TO. Lack of association of the esophageal microbiome in adults with eosinophilic esophagitis compared with non-EoE controls. J Gastrointestin Liver Dis. 2021 Mar 12;30(1):17-24. doi: 10.15403/jgld-3049. PMID: 33723541.
- The authors should include this work in discussion of microbiome in EoE
Harris JK, Fang R, Wagner BD, Choe HN, Kelly CJ, Schroeder S, Moore W, Stevens MJ, Yeckes A, Amsden K, Kagalwalla AF, Zalewski A, Hirano I, Gonsalves N, Henry LN, Masterson JC, Robertson CE, Leung DY, Pace NR, Ackerman SJ, Furuta GT, Fillon SA. Esophageal microbiome in eosinophilic esophagitis. PLoS One. 2015 May 28;10(5):e0128346. doi: 10.1371/journal.pone.0128346. PMID: 26020633; PMCID: PMC4447451.
- In the conclusion, in line 352, this should be reworded as it is not a "study". Would recommend to change to "Here we performed a literature review to summarize evidence on potential non-invasive..."
Quality of english is good. There are a few minor typos to review.
Author Response
Reviewer #1
C0. The authors wrote a nice review on biomarkers in EoE, going in depth on previous studies evaluate serum, esophageal, and fecal biomarkers.
R0. We thank the reviewer for her/his positive evaluation of our work.
C1. While I agree that minimally invasive biomarkers would advance the field, the authors state in the abstract and introduction EGDS are "often poorly tolerated" which is not universally accepted. The authors should point out that adverse effects of EGDS are considered rare both in adult and pediatric literature and cite this. In addition the adverse events are often related to anesthesia and not necessarily EGD.
R1. We thank the reviewer for her/his important comment. In the abstract and introduction, we specified that endoscopy without sedation is often poorly tolerated.
C2. The authors do a nice job of outlining each biomarker with data from serum, esophageal eluate and feces.
R2. We thank the reviewer for her/his positive comment on our work.
C3. The authors should consider including a paragraph / section about other less invasive modalities including cytosponge, blind esophageal brushing, esophageal string test, unsedated transnasal endoscopy.
Katzka DA, Smyrk TC, Alexander JA, Geno DM, Beitia RA, Chang AO, Shaheen NJ, Fitzgerald RC, Dellon ES. Accuracy and Safety of the Cytosponge for Assessing Histologic Activity in Eosinophilic Esophagitis: A Two-Center Study. Am J Gastroenterol. 2017 Oct;112(10):1538-1544. doi: 10.1038/ajg.2017.244. Epub 2017 Aug 15. Erratum in: Am J Gastroenterol. 2017 Dec 19;: PMID: 28809387; PMCID: PMC5848207.
Katzka DA, Geno DM, Ravi A, Smyrk TC, Lao-Sirieix P, Miremadi A, Debiram I, O'Donovan M, Kita H, Kephart GM, Kryzer LA, Camilleri M, Alexander JA, Fitzgerald RC. Accuracy, safety, and tolerability of tissue collection by Cytosponge vs endoscopy for evaluation of eosinophilic esophagitis. Clin Gastroenterol Hepatol. 2015 Jan;13(1):77-83.e2. doi: 10.1016/j.cgh.2014.06.026. Epub 2014 Jul 3. Erratum in: Clin Gastroenterol Hepatol. 2015 Aug;13(8):1552. Miramedi, Ahmed [corrected to Miremadi, Ahmed]. PMID: 24997328.
Smadi Y, Deb C, Bornstein J, Safder S, Horvath K, Mehta D. Blind esophageal brushing offers a safe and accurate method to monitor inflammation in children and young adults with eosinophilic esophagitis. Dis Esophagus. 2018 Dec 1;31(12). doi: 10.1093/dote/doy056. PMID: 29905784.
Nguyen N, Lavery WJ, Capocelli KE, Smith C, DeBoer EM, Deterding R, Prager JD, Leinwand K, Kobak GE, Kramer RE, Menard-Katcher C, Furuta GT, Atkins D, Fleischer D, Greenhawt M, Friedlander JA. Transnasal Endoscopy in Unsedated Children With Eosinophilic Esophagitis Using Virtual Reality Video Goggles. Clin Gastroenterol Hepatol. 2019 Nov;17(12):2455-2462. doi: 10.1016/j.cgh.2019.01.023. Epub 2019 Jan 29. PMID: 30708107; PMCID: PMC6663663.
Philpott H, Nandurkar S, Royce SG, Gibson PR. Ultrathin unsedated transnasal gastroscopy in monitoring eosinophilic esophagitis. J Gastroenterol Hepatol. 2016 Mar;31(3):590-4. doi: 10.1111/jgh.13173. PMID: 26426817.
R3. We thank the reviewer for her/his comment. The aim of this narrative review was to summarize eosinophil and non-eosinophil-derived biomarkers other than eosinophil counts. In this regard, there are no studies that investigated EoE biomarkers other than eosinophil counts with Cytosponge or transnasal endoscopy. With regards to blind esophageal brushing and esophageal string test, these two modalities are discussed throughout the manuscript. We already included the reference of Smadi et al. that you suggested on the blind esophageal brushing (ref #55). While references #28 and #41 are of studies using the esophageal string test. In addition, to address your comment, we discussed transnasal endoscopy and Cytosponge studies in the Conclusion section and referenced the studies you suggested.
C4. The authors review each biomarker obtained from the esophageal string test under each heading, but they should include overall description of the esophageal string test with EoE Score (which specifically uses eotaxin-3 and major basic protein-1 concentrations) that distinguished subjects with active EoE from inactive EoE or normal. The EST is now being used clinically and not for research purposes only at several institutions and this should be noted.
R4. We thank the reviewer for her/his relevant comment. We modified the manuscript as suggested and described the EST-based EoE score. Please see page #7 lines 283-287.
C5. The biomarkers being reviewed overall alone are not being used in clinical practice as noted, but the authors should highlight less invasive modalities that are being used (Esophageal string test and transnasal endoscopy).
R5. We thank the reviewer for her/his comment. To address the comment we discussed other less-invasive monitoring strategies in the conclusion section.
C6. In Section 7, line 301, the authors delve into recent literature on IgG4. This should be expanded to why IgG4 is believed to be involved in EoE so the reader understands why this is being evaluated as a biomarker.
R6. We thank the reviewer for her/his appreciated comment. We added information regarding IgG4 in EoE.
C7. In line 319, the authors discuss the microbiome. They do not discuss the studies that show lack of association of microbiome and EoE.
Johnson J, Dellon ES, McCoy AN, Sun S, Jensen ET, Fodor AA, Keku TO. Lack of association of the esophageal microbiome in adults with eosinophilic esophagitis compared with non-EoE controls. J Gastrointestin Liver Dis. 2021 Mar 12;30(1):17-24. doi: 10.15403/jgld-3049. PMID: 33723541.
R7. We thank the reviewer for her/his suggestion. We discussed this study as suggested.
C8. The authors should include this work in discussion of microbiome in EoE
Harris JK, Fang R, Wagner BD, Choe HN, Kelly CJ, Schroeder S, Moore W, Stevens MJ, Yeckes A, Amsden K, Kagalwalla AF, Zalewski A, Hirano I, Gonsalves N, Henry LN, Masterson JC, Robertson CE, Leung DY, Pace NR, Ackerman SJ, Furuta GT, Fillon SA. Esophageal microbiome in eosinophilic esophagitis. PLoS One. 2015 May 28;10(5):e0128346. doi: 10.1371/journal.pone.0128346. PMID: 26020633; PMCID: PMC4447451.
R8. We thank the reviewer for her/his suggestion. We discussed this study as suggested.
C9. In the conclusion, in line 352, this should be reworded as it is not a "study". Would recommend to change to "Here we performed a literature review to summarize evidence on potential non-invasive..."
R9. We thank the reviewer for her/his point on this. We agree and reworded as suggested.

Reviewer 2 Report
The article raises an important clinical problem, which is the diagnosis and monitoring of EoE. As the authors noted, the incidence of EoE in the last years of the disease, and the costs of social care from EoE, including specific costs from other chronic diseases, e.g. inflammatory bowel disease. We have effective treatments for EoE, that we start after diagnosis of the disease is made and/or lack of response to other therapies is confirmed.
We suspect the disease in a patient with dysphagia, food impaction as well as in patients with GERD refractory do PPIs. Diagnosis of EoE requires gastroscopy with esophageal biopsy. Although endoscopy is widely available, there are problems in diagnosing EoE. First, several specimens should be taken for histopathological examination in case of EoE suspicion. Nevertheless, there are difficulties in the diagnosis of EoE and sometimes the examination needs to be repeated. The recent (2022) ACG guidelines for the management of GERD recommend gastroscopy with biopsy after discontinuation of PPIs for 2 to 4 weeks to increase the diagnostic yield (please consider to add this comment to the Introduction, ref. Am J Gastroenterol. 2022 Jan 1; 117(1): 27–56). PPIs can eliminate the endoscopic and histologic features of EoE. The diagnosis of EoE cannot be excluded if endoscopy is performed while the patient is taking PPIs (please consider to add this comment to the Introduction, ref. Gastroenterology, 2018. 154(5): p. 1217–1221.e3).
In addition, it is a chronic disease and patients need to be monitored. In order to assess the effectiveness of therapy and its modification, it is necessary to perform endoscopy with biopsy. Finding non-invasive or minimally invasive sensitive and specific markers that facilitate diagnosis and treatment monitoring is a priority and will allow to reduce the number of endoscopies in patients. The article discusses the current knowledge on several known markers such as: ECP, EDN and MBP as well as other non-promising-eosinophil-derivered biomarkers. The article indicated the possibilities of their identification and limitations as well as future directions of research. Currently, none of these markers are used in everyday clinical practice. Please indicate clinical situations where we can observe increased levels of ECP, EDN, and MBP (e.g. rhinitis, asthma, atopic dermatitis, melanoma).
Further research is needed to validate their usefulness in the diagnosis and follow-up of EoE.
Author Response
Reviewer #2
C0. The article raises an important clinical problem, which is the diagnosis and monitoring of EoE.
R0. We thank the reviewer for her/his positive evaluation on our work.
C1. As the authors noted, the incidence of EoE in the last years of the disease, and the costs of social care from EoE, including specific costs from other chronic diseases, e.g. inflammatory bowel disease. We have effective treatments for EoE, that we start after diagnosis of the disease is made and/or lack of response to other therapies is confirmed. We suspect the disease in a patient with dysphagia, food impaction as well as in patients with GERD refractory do PPIs. Diagnosis of EoE requires gastroscopy with esophageal biopsy. Although endoscopy is widely available, there are problems in diagnosing EoE. First, several specimens should be taken for histopathological examination in case of EoE suspicion. Nevertheless, there are difficulties in the diagnosis of EoE and sometimes the examination needs to be repeated. The recent (2022) ACG guidelines for the management of GERD recommend gastroscopy with biopsy after discontinuation of PPIs for 2 to 4 weeks to increase the diagnostic yield (please consider to add this comment to the Introduction, ref. Am J Gastroenterol. 2022 Jan 1; 117(1): 27–56). PPIs can eliminate the endoscopic and histologic features of EoE. The diagnosis of EoE cannot be excluded if endoscopy is performed while the patient is taking PPIs (please consider to add this comment to the Introduction, ref. Gastroenterology, 2018. 154(5): p. 1217–1221.e3).
R1. We thank the reviewer for her/his comment. We added the references of suggested studies.
C2. In addition, it is a chronic disease and patients need to be monitored. In order to assess the effectiveness of therapy and its modification, it is necessary to perform endoscopy with biopsy. Finding non-invasive or minimally invasive sensitive and specific markers that facilitate diagnosis and treatment monitoring is a priority and will allow to reduce the number of endoscopies in patients. The article discusses the current knowledge on several known markers such as: ECP, EDN and MBP as well as other non-promising-eosinophil-derivered biomarkers. The article indicated the possibilities of their identification and limitations as well as future directions of research. Currently, none of these markers are used in everyday clinical practice. Please indicate clinical situations where we can observe increased levels of ECP, EDN, and MBP (e.g. rhinitis, asthma, atopic dermatitis, melanoma).Further research is needed to validate their usefulness in the diagnosis and follow-up of EoE.
R2. We thank the reviewer for her/his important comment. We modified the manuscript according to the reviewer’s suggestion and specified clinical situations with increased levels of eosinophil-derived biomarkers. Please see Section 2, page #2, lines 81-82

Round 2
Reviewer 1 Report
Thank you for making edits to the manuscript based on previous comments.